# Recycled Porcine Bone Powder as Filler in Thermoplastic Composite Materials Enriched with Chitosan for a Bone Scaffold Application

**DOI:** 10.3390/polym13162751

**Published:** 2021-08-16

**Authors:** Marco Valente, Jordi Puiggalí, Luis J. del Valle, Gioconda Titolo, Matteo Sambucci

**Affiliations:** 1Department of Chemical Engineering, Materials, Environment, Sapienza University of Rome, 00184 Rome, Italy; titologioconda@gmail.com (G.T.); matteo.sambucci@uniroma1.it (M.S.); 2INSTM Reference Laboratory for Engineering of Surface Treatments, Department of Chemical Engineering, Materials, Environment, Sapienza University of Rome, 00184 Rome, Italy; 3Departament d’Enginyeria Química, Universitat Politècnica de Catalunya, Escola d’Enginyeria de Barcelona Est-EEBE, 08019 Barcelona, Spain; jordi.puiggali@upc.edu (J.P.); luis.javier.del.valle@upc.edu (L.J.d.V.)

**Keywords:** thermoplastic composites, bone scaffold, recycled bone powder, chitosan, PLA, PCL, mechanical properties, antibacterial activity

## Abstract

This work aims to synthesize biocompatible composite materials loaded with recycled porcine bone powder (BP) to fabricate scaffolds for in-situ reconstruction of bone structures. Polylactic acid (PLA) and poly(ε-caprolactone) (PCL) were tested as matrices in percentages from 40 wt% to 80 wt%. Chitosan (CS) was selected for its antibacterial properties, in the amount from 5 wt% to 15 wt%, and BP from 20 wt% to 50 wt% as active filler to promote osseointegration. In this preliminary investigation, samples have been produced by solvent casting to introduce the highest possible percentage of fillers. PCL has been chosen as a matrix due to its greater ability to incorporate fillers, ensuring their adequate dispersion and lower working temperatures compared to PLA. Tensile tests demonstrated strength properties (6–10 MPa) suitable for hard tissue engineering applications. Based on the different findings (integration of PLA in the composite system, improvements in CS adhesion and mechanical properties), the authors supposed an optimization of the synthesis process, focused on the possible implementation of the electrospinning technique to develop PCL-BP composites reinforced with PLA-CS microfibers. Finally, biological tests were conducted to evaluate the antibacterial activity of CS, demonstrating the applicability of the materials for the biomedical field.

## 1. Introduction

Continuous advances in medical science and surgical techniques have allowed transplantation, whether of tissues or whole organs, to become one of the potential options for restoring the native functions of many damaged parts of the human body. However, transplantation techniques encountered several limitations, such as the growing demand that far exceeds the actual availability by donors of usable tissues and the possible contamination of the donor tissue [1]. Therefore, a new technique is needed to reduce this discrepancy between clinical needs and availability in healthy tissues or organs. Tissue engineering (TE) is an interdisciplinary approach which utilizes cells, bioactive factors, and biomaterials to restore required functions of tissues. TE is based on the principles of medicine, biology, material science, and engineering by integrating them into the design of biological components to repair, maintain, and improve tissue functions [2,3]. Scaffolds are an essential part of this methodology. They are 3D structures that provide a template for cell attachment in the biological tissue formation. Besides, scaffolds perform unique functions that cannot be achieved by common drugs, including providing specific mechanical support in defect areas, stimulating cell differentiation by regulating the mechanical properties, and executing, raising, or replacing some lost physiological functions caused by diseases or impairments [2,4]. In summary, an ideal scaffold in TE applications should meet three basic requirements: (a) A porous microstructure with a tunable pore size distribution; (b) biocompatibility and degradation properties to enable cell migration and growth; (c) appropriate mechanical properties and stability of the shape to resist stresses and maintain the integrity of the designed structure [5]. According to the classification proposed by Eltom et al. [6], fabrication technologies for scaffold manufacturing can be categorized in two classes: Conventional fabrication techniques (freeze-drying, solvent casting, gas foaming, electrospinning) and rapid prototyping methods (stereolithography, selective laser sintering, fused deposition modelling, bioprinting). Table 1 shows the main advantages and limitations of these scaffold fabrication technologies compared with the transplantation approach applied in tissue surgery.

In the orthopedical sector, bone tissue engineering (BTE) is a rapidly growing sector aiming to create a bio-functional tissue that can integrate and degrade in vivo to treat lost, diseased, or damaged bone tissue, minimizing the complications deriving from the more traditional bone grafting methods [10]. According to the principles of TE, BTE is based on four key components: (a) Osteogenic cells that generate the bone matrix; (b) a scaffold that mimics the extracellular matrix (ECM) of the bone; (c) vascularization that ensures the transport of nutrient and waste products; and (d) morphogenetic signals to assist cell proliferation. Osteo-induction, osteo-conduction, osseointegration, chemical-mechanical stability in host environment, and adequate manufacturability are the main requirements that a biomaterial should possess for BTE applications [11]. In this context, several matrices have been identified as functional for scaffold applications due to their suitable mechanical and technological properties. Specifically, Polylactic acid (PLA) and poly(ε-caprolactone) (PCL) were recognized as promising candidates in the BTE field. They are biocompatible and biodegradable polymers whose degradation products (due to non-enzymatic hydrolysis of the esters in a physiological environment) are low molecular weight compounds, such as lactic and hydroxyhexanoic acids [12,13]. For instance, lactic acid is the main degradation product that falls within the normal metabolic pathways. In fact, it is normally expelled by the body in the form of carbon dioxide and water [12,13]. The ability of the two thermoplastic polymers to support osseointegration without undergoing rejection from the biological environment was demonstrated in recent studies. Nazeer et al. [14] researched 3D-printed PLA scaffolds functionalized with chitosan (CS) and hydroxyapatite (HA) used as mechanical reinforcements and an osteoconductive filler, respectively. Zimina et al. [15] employed solvent casting and salt leaching methods to fabricate PLA/HA scaffolds with mechanical properties suitable for soft TE applications and minimal inflammatory response after in vivo implantation. Lopresti et al. [16] investigated the influence of HA concentration and size on the mechanical properties and cytocompatibility of PLA-based elctrospun nanofibrous mats intended for BTE applications. The nanosized HA fillers promoted less structural defects, higher mechanical performance, and better biological behavior in terms of more homogeneous colonization of the scaffold by pre-osteoblastic cells. Trakoolwannachai et al. [17] synthesized HA from eggshell waste to produce PCL-HA composites by a conventional wet chemical method. In-vitro analysis showed that the scaffolds supported high levels of Saos-2 cells without causing any cytotoxic effect. Cestari et al. [18] developed PCL scaffolds, via 3D printing, functionalized with nanometric HA. The composite scaffolds showed a compressive elastic modulus between 203 MPa and 316 MPa, comparable with that of trabecular bone, and better in vitro bioactivity than pure PCL. Miszuk et al. [19] synthesized a 3D electrospun PCL/HA scaffold combined with biomaterials-mediated bone morphogenic proteins (BMPs) to generate a bone-forming favorable niche for BTE interventions. By in vivo and in vitro experimentation, the developed scaffolds (~96% of interconnected porosity) were able to form a system to generate more favorable microenvironments for osteogenic differentiation and BMP2-induced ectopic bone formation than plain 3D electrospun PCL scaffolds. From the previously cited literature, two noteworthy aspects can be deduced: (a) The high adaptability of the two thermoplastic polymers to be processed with a wide range of TE manufacturing methods and (b) the use of HA as a bioactive, osteoconductive, and mechanical reinforcement filler.

HA is a bioactive and bioresorbable Calcium phosphate that constitutes most of the inorganic components of bone tissue. In both natural and synthetic form, HA has an identical chemical structure that differs in terms of porosity, crystal size, and microstructure. HA directly bonds with live bone after implantation in cases of bone defects, enhancing appropriate vascularization, stem cell proliferation, and the bone regeneration without causing any local or systemic toxicity [20]. There are two common ways to obtain HA [21]: Synthetically (wet processing, dry processing, sol-gel, hydrothermal processing) or by extraction from naturally available materials (eggshells, fish scales, bovine bones). The synthetic product is still preferred for BTE applications thanks to its higher purity and biological and mechanical effectiveness [22]. However, several drawbacks were found in its employment [21,22,23], such as the use of expensive regent chemicals, the involvement of time-consuming and laborious work, possible triggering of inflammatory reactions by the host system due to implanted debris or foreign substances, and the slow degradation rate that hindered the rapid osteogenesis.

To investigate alternative osteo-active fillers for the BTE field, the present work proposed a preliminary study about the possibility of integrating recycled bone powder (BP) in thermoplastic matrices (PLA and PCL), to develop biocompatible thermoplastic composite materials suitable for scaffold applications. This research is positioned in the novel perspective of industrial waste recovery to obtain new materials for TE, understanding the importance of recycling of waste materials and by-products to obtain alternative source for precursors. Indeed, BP used like an inorganic reinforcement was a fine-grain (0–500 μm) by-product of the mechanical processing of porcine bone to obtain two coarser bone fractions used by Tecnoss™ company (Turin, Italy) [24] in OsteoBiol™ technology (Figure 1).

Besides, the research envisaged an optimization in terms of antimicrobial properties, by the integration of Chitosan (CS), extensively used as a “platform” in BTE and other biomedical applications because of its cytocompatibility, degradability, appropriateness for cell ingrowths, and bactericidal activity [25]. As a first attempt at manufacturing the composite, the solvent-casting technique was employed to produce bio-composites. The aim of the research was to investigate the feasibility of producing composite materials with adequate reproducibility and analyze the variations in mechanical and antibacterial properties. After evaluating the most suitable polymer-BP-CS formulations, mechanical characterization by tensile tests and bacterial growth/adhesion evaluation were performed. This preliminary study lays the foundations for advanced upgrades in terms of integration of reinforcement fillers and the development of filaments potentially usable in additive manufacturing (AM) techniques.

## 2. Materials and Methods

### 2.1. Materials

PLA Ingeo Biopolymer 3001D (density of 1.24 g/cm^3^ at 25 °C; melt flow rate of 80 g/10 min at 210 °C; glass transition temperature of 55–60 °C) was purchased from NatureWorks (Minnetonka, MN, USA). PCL pellets (molecular weight of 80,000 g/mol; density of 1.145 g/mL; melting point of 60 °C; purity > 95%), CS (high molecular weight of 310–357 kDa, with a degree of deacetylation > 75%, viscosity of 800–2000 cps, density of 0.15–0.3 g/cm^3^), and 99% anhydrous Chloroform (CHCl_3_) stabilized with amylene were purchased from Sigma-Aldrich (St. Louis, MO, USA). As previously mentioned, 0–500 μm porcine BP was provided by Tecnoss (Turin, Italy). The microstructure (Figure 2) and particle size distribution (Figure 3) of the particles were investigated by scanning electron microscopy (SEM), using a Mira 3 FEG-SEM (Tescan, Brno, Czech Republic), and particle size analysis by a MasterSizer 3000 (Malvern Panalytical, Malvern, UK), respectively.

### 2.2. Experimental

#### 2.2.1. Solvent Casting

BP and CS were added to the appropriate solutions of PCL and PLA in CHCl_3_. Mixtures were then homogenized by stirring for 24 h inside a closed shaker at 37 °C and 95 rpm. The final mixtures were finally poured into a glass Petri dish with a diameter of 9 cm and height of 1.5 cm, to obtain films with a maximum thickness of 1 mm after complete solvent evaporation. The solvent-casting process lasted 4 days. In accordance with the above procedure, different formulations were tested by varying the type of matrix (PCL and PLA) and the percentage of fillers (CS and BP). For each thermoplastic matrix, four weight levels of BP (from 20 wt% to 50 wt%) and three weight percentages of CS (from 5 wt% to 15 wt%) were selected and experimented with.

After the process, the solvent-casted products were subjected to quality and dimensional checks, to evaluate the characteristics of the film in terms of structural integrity and thickness homogeneity. The composite films had to be uniform, not excessively brittle, and able to incorporate BP inside. The dimensional check was performed by a MEGA-CHECK Pocket (List-Magnetik, Leinfelden-Echterdingen, Germany) thickness meter (Figure 4), taking into consideration only the films with a thickness in the central part of 0.8–1.2 mm, which appeared to be suitable for the preparation of specimens for mechanical tests.

#### 2.2.2. Samples Preparation for Mechanical Tests

Specimens for tensile tests were obtained from the solvent-casted films by die-cutting (Figure 5), using a MT1130 die-cutter (MinEuro, Turin, Italy) equipped with a metallic dog-bone shaped cutting edge. From each disk, it was possible to extract three samples for the mechanical characterization, with a width of 3 mm and a gauge length of 60 mm.

### 2.3. Characterization

#### 2.3.1. SEM

The microstructural evaluation of the samples by SEM was conducted in collaboration with the Department of Chemical Engineering, Materials, Environment (Sapienza University of Rome, Italy) and Department d’Enginyeria Química (Universitat Politècnica de Catalunya, Barcelona, Spain) employing a Mira 3 FEG-SEM (Tescan, Brno, Czech Republic) and a Phenom XL Desktop SEM (Thermo-Fisher, Waltham, MA, USA), respectively. Before SEM analyses, samples were sputter-coated with gold to obtain conductive specimens.

#### 2.3.2. Mechanical Tests

Tensile tests were performed at room temperature using a ZwichRoell Z10 machine (Zwick-Roell, Ulm, Germany) equipped with a 1kN load cell and interfaced with the TESTXPERT II (Zwick-Roell, Ulm, Germany) software. The crosshead speed was 5 mm/min.

#### 2.3.3. Assay of Bacterial Growth and Adhesion

*Staphylococcus aureus* ATCC 25923 (*S. aureus)* and *Escherichia coli* DH5α (*E. coli)* bacteria were selected to evaluate the antimicrobial activity of CS in the PLA matrices. Bacterial inhibition was quantitatively evaluated.

The samples (i.e., each one of 100 mg) were placed into the wells of a 24-well culture plate and sterilized by UV radiation in a laminar flux cabinet for 15 min. Then, 1 mL of broth culture (Luria-Bertani medium) containing the bacteria at a concentration of 10^3^ CFU/mL was added to samples for quantitative tests. Cultures were incubated at 37 °C and agitated at 80 rpm. Aliquots of 50 μL were taken at predetermined time intervals (i.e., time 0 and 24 h) for absorbance measurement at 650 nm in a microplate reader. Each sample was analyzed in quadruplicate and the results averaged. *t*-Test with a 95% (*p* < 0.05) confidence level was performed to determine significant differences of the samples. Turbidity was related to the relative bacterial growth by considering the maximum growth attained in the control, the sample without CS.

Bacterial adhesion on matrices has also been determined. The culture medium was aspirated after the proliferation measurements (i.e., at 72 h of culture), and the material was washed three times with distilled water. Then, 0.5 mL of sterile 0.01 M Sodium thiosulfate was added to each well to detach adhered bacteria on the sample surface. After agitation at 100 rpm and 37 °C for 1 h, samples were removed, and 1 mL of broth culture was added to each well. The first sample was taken subsequently filled with fresh broth culture (time 0 h for adhesion assay). Then, plates were incubated at 37 °C and 100 rpm for 24 h. The bacterial number was determined by absorbance measurement at 24 h.

## 3. Results and Discussions

### 3.1. Quality Check of Post-Solvent-Casted Specimens

Only some PCL-based formulations were suitable to produce suitable solvent-casted disks and dog-bone specimens for mechanical tests. PLA-based composite samples were excessively curved, too inhomogeneous, or unable to incorporate the inorganic reinforcement. In this regard, Zhou et al. [26] highlighted that one of the main concerns in the use of solvent casting techniques to integrate HA fillers in the PLA matrix concerns the low affinity between the two components: During the solvent evaporation, the particles can precipitate from the polymer solution, resulting in a non-uniform dispersion or weak incorporation. By way of example, the comparison between a PCL-based disk (30 wt% of BP) with good dimensional and mechanical characteristics and an untestable PLA-based sample (50 wt% of BP) is shown in Figure 6. This variability of results is attributable to a non-optimized solvent-casting process. A possible future development will be based on the careful study of the synthesis parameters (the polymer–solvent ratio, solvent evaporation conditions, and the mixing method) and integrating alternative technologies to incorporate BP and CS into the polymer matrix. This latter aspect will be discussed in more detail later.

Table 2 lists the PCL formulations (plain and composite specimens) subjected to mechanical characterization. Here, 5 wt% was the maximum level of CS that was possible to incorporate in the composite, maintaining suitable dimensional and structural characteristics of solvent-casted films for the dog-bone specimen extraction and adequate dispersion and incorporation of the fillers into the thermoplastic matrix.

### 3.2. Microstructural Characterization by SEM

#### 3.2.1. PLA-Based Samples Prepared by Solvent Casting

SEM analysis on samples extracted from PLA disks confirms the defects detected during the post-solvent-casting quality control. In Figure 7a, the cross section of the PLA-based composite (40 wt% of BP) shows an inhomogeneous distribution of the BP, which appears totally absent in the central part of the analyzed fragment. In addition, some small cracks are detectable. The high-magnification micrograph (Figure 7b) displays the defects, for example due to air bubbles incorporated in the solution during stirring and casting in the Petri dish. In the same figure, areas with poor solvent evaporation are evident. Figure 7c shows the air-exposed surface of the PLS sample incorporated with 50 wt% of BP. In this case, a completely polymeric surface emerges without any presence of BP. A poor interface between PLA and BP was observed in Figure 7d. This evidence agrees with the literature data [27]. The poor interface leads to a collapse of the mechanical properties of the sample, due to the absence of an interaction between the matrix and filler. Therefore, these matrices may not be ideal scaffolds, and, as mentioned above, improvements in their preparation would be required.

#### 3.2.2. PCL-Based Samples Prepared by Solvent Casting

Unlike the PLA-BP composites, the solvent-casting procedure employed in this research appeared to be quite efficient for PCL matrix. From the cross-section image of PCL50-BP50 fragment, it can be seen how the polymer incorporates BP, which is homogeneously distributed over the investigated section (Figure 8a). Although the distribution of the ceramic fillers in the matrix was adequate, weak matrix–particle adhesion was noted (Figure 8b). The cross-section micrograph in Figure 8c highlights the distribution of CS filler (dashed areas) in the PCL45-BP50-CS5 sample. Figure 8d details the typical flake-like shape of CS and its interface with the PCL matrix. The tensile test results described in the next section will provide an assessment on the impact of the microstructural features on the mechanical properties.

### 3.3. Mechanical Characterization by Tensile Tests

Tensile tests were performed on the five PCL-BP-CS composites described in Table 1 and the results are summarized by histograms in Figure 9 (Tensile strength results), plots in Figure 10 (Young’s modulus and elongation-at-break results), and stress–strain curves in Figure 11.

Although SEM investigation on PCL-BP composites revealed some defects in matrix–ceramic interfaces, the tensile strength of the samples did not show significant drops compared to unfilled material (PCL-0). These results confirm that PCL and BP in this proportion are compatible in terms of dispersion. Tensile strength increased from 9.34 MPa to 10.32 MPa when increasing the loading of BP from 0% to 50%, indicating an effective reinforcement effect of hard-phase fillers. The most significant decrease in mechanical strength is observed in the formulation functionalized with 5 wt% of CS fillers. This behavior is attributable to several possible reasons [27,28]: (a) The flaked morphology of CS reduces the interlocking with the PCL matrix; (b) the strong difference in polarity between PCL and CS results in phase-separated composites, reducing the mechanical properties; and (c) the non-optimal concentration of CS, which could lead to an agglomeration phenomena and scarce interfacial adhesion with the matrix. As reported by [29], for hard tissue engineering applications, a minimum strength value of 100 kPa is required (depending on target tissue). All the formulations developed in this research fully satisfy such requirements.

By increasing the BP content, the material’s plasticity was decreased while stiffness increased. For BP levels below 30 wt%, the composites maintain comparable elasticity properties, indicating that the polymer matrix is in such quantities as to still govern the mechanical behavior of the material. Above this concentration, there is a sudden increase in the material’s stiffness. The maximum Young’s modulus value can be found in the PCL45-BP50-CS5 sample, where an increase of about 385% compared to the plain formulation (PCL-0) was recorded. In the CS-filled composite, the increase in stiffness did not correspond to an increase in tensile strength as usually observed, but a marked mechanical strength drop was detected. In addition to the factors mentioned above regarding the CS–PCL interaction, this behavior can be attributed to the lower PCL concentration more so than the other formulations. A similar finding was found by Fadaie et al. [30]: Reducing the concentration of the PCL solution causes a remarkable decline in tensile strength. Elongation-at-break gradually decreases with the increase of BP more incisively than the elastic modulus increase previously observed. For a low BP level (20 wt%) there is already a strong decrease in deformability (−54% of elongation-at-break) and a more brittle failure behavior compared to the unloaded PCL sample. Therefore, in accordance with previous studies [31,32], the co-presence of ductile polymer and hard ceramic fillers promotes the brittle nature of the bio-composite.

### 3.4. Process Upgrade

In light of the results obtained and discussed in the previous sections, this part anticipates a possible implementation of the production technology of thermoplastic-ceramic composites. Specifically, it illustrates a way to ensure the integration of PLA into the system by overcoming the problems encountered in the solvent-casting method. The idea is based on the use of the electrospinning technique to obtain PLA microfibers loaded with CS to be deposited on the PCL-BP disk. This would allow one to minimize the CS–matrix interface issues and ensure greater mechanical strength of the composite due to the fiber-reinforcing effect. A PCL disk reinforced with electrospun PLA-CS microfibers can be reduced to pellets for subsequent extrusion, obtaining the PCL-PLA-BP-CS filament. In this case, the great advantage concerns the correct balance in the physical properties of the two biopolymers: PCL melts at about 90 °C while PLA melts at 150–160 °C. This would allow the polymeric fibers blended with CS to be kept unchanged during the extrusion, resulting in a reinforced filament with adequate fiber alignment and the proper incorporation of BP and CS fillers. A schematic illustration of the proposed production cycle is shown in Figure 12.

Based on this possible implementation of the production technology, the biological investigations described below focused on the PLA-based formulations, evaluating the antibacterial efficiency of CS. Two PLA-BP-CS composites (Table 3), previously processed by solvent casting, were analyzed to evaluate the influence of different concentration of CS on the antibacterial performance.

### 3.5. Antibacterial Activity of PLA-BP-CS

CS is largely known for its activity against a wide range of microorganisms. However, its antibacterial activity can be related to its structural characteristics. As reviewed by Chandrasekaran et al. [33], the antibacterial properties of low-molecular weight (MW) CS result in being higher or lower than those of high MW CS depending on the unit of measure with which concentrations are expressed. If concentrations are expressed in µg/mL, high MW CS usually results in being less potent than low molecular weight CS. Conversely, if the concentrations are expressed as µM, high MW CS results in being more potent than low molecular weight CS. Past studies evaluated the antibacterial activity of high MW CS (similar to that used in this research work) against Gram-negative and Gram-positive bacteria. No et al. [34] demonstrated a stronger bactericidal effect for Gram-positive species than Gram-negative ones, evaluating a reduction of bacterial activities of 36% and 82% against *E. coli* and *S. aureus*, respectively. Mellegard et al. [35] observed that low MW CS (below 12 kDa) had no detectable inhibitory effect in terms of antibacterial activities towards both Gram-negative and Gram-positive cultures. On the other hand, high MW CS showed greater antibacterial activity. Specifically, the bactericidal effect, evaluated in terms of minimum bactericidal concentration (MBC), was higher against Gram-negative species (*E. coli*) than the Gram-negative strain (*B. cereus*), recording MBC values of 0.13 mg/mL and 0.95 mg/mL, respectively.

In this study, the bactericide effect of CS loaded in the matrix was determined by studying bacterial adhesion on the matrix and the bacterial growth in broth medium. As shown in Figure 13, regardless of the type of bacterial culture, the PLA60-BP25-CS15 sample promoted lower bacterial adhesion than PLA65-BP30-CS5. A higher CS concentration achieved its antimicrobial mechanism, based on the presence of charged groups in the polymer backbone and their ionic interactions with bacteria wall constituents. This interaction suggests the occurrence of hydrolysis of the peptidoglycans in the bacterial wall, provoking the leakage of intracellular electrolytes, leading the bacteria to death. The charges present in CS chains are generated by the protonation of amino groups when in an acid medium [36]. Globally, a lower bacterial adhesion rate was observed for *S. aureus* bacteria with respect to the *E. coli* strain. This finding agrees with previous studies [37,38], which revealed that CS is much more incisive on Gram-positive bacteria (i.e., S. *aureus*) than Gram negative ones (i.e., *E. coli*). Although it is widely accepted that CS inhibits bacterial growth through an electrostatic interaction between its protonated amino groups and negative residues on microbial cell membranes, it is controversial whether this activity can be increased by lowering or increasing the MW [39].

Bacterial growth measurements (Figure 14) further proved the bactericide activity of CS, indicating a greater efficiency for a higher filler load (i.e., 15% vs. 4% of CS). Comparing the investigated bacterial cultures, the analysis on *E. coli* strain revealed a maximum reduction of bacterial growth of about 9%, while more relevant reduction was observed in the case of *S. aureus* bacteria (maximum reduction rate of about 16%).

The results obtained allow us to conclude that the CS load in the matrices prepared for bone regeneration maintain its antibacterial activity and the potency of this activity depends on its load ratio in the matrix. The presence of CS inhibits the adhesion of bacteria on the material and the release of CS from the matrix can control bacterial growth in the matrix environment. Better performance was found against the Gram-positive bacterial culture (*S. aureus*), which was recognized as a high-incidence factor for hospital infections [40].

## 4. Conclusions

This research work presented a preliminary investigation on the synthesis of thermoplastic composites loaded with waste BP and CS filler, to obtain bioactive materials for BTE applications. PCL-BP and PLA-BP composites (at different BP content) were processed in the form of solvent-casted films to evaluate the ability of thermoplastic matrices to incorporate the inorganic fillers and obtain homogeneous and workable composites. From a qualitative check of the films and SEM investigations, it was found that only PCL ensured an adequate incorporation of the BP and good particle dispersion. Conversely, in the case of PLA, weak structural integrity of the films and poor integration of the filler was detected. Even in relation to the lowest working temperatures compared to PLA, PCL was selected as the matrix for the bio-composite under study. Mechanical tests were performed on the PCL-BP-CS formulations to evaluate the effect of the two fillers on the mechanical performance of the material. Although several BP-PCL interfacial defects were observed, the formulations investigated show suitable mechanical strength properties for tissue engineering applications. However, a drastic decline in mechanical properties was found when the CS was incorporated into the composite, resulting in the chemical-physical incompatibility between PCL and CS. According to these findings, the authors hypothesized a proposal for the optimization of the production technology based on the integration of the electrospinning technique to fiber-reinforce the PCL-BP composites with PLA-CS micro-fibers, in order to minimize the integration problems of the PLA in the composite, promote the adhesion of CS, and obtain a potential mechanical strength improvement. Based on this upgrade, antibacterial tests were performed on the PLA-BP-CS composites, which highlighted the effective bactericidal efficiency of the CS and the potential applicability of the composite for biomedical applications.

Future research will be aimed at investigating the optimized process proposed by the authors, firstly evaluating the possibility of implementing electrospinning technology in the production of PCL-PLA-BP-CS composites. An experimental campaign focused on the production of the fiber-reinforced composite filaments will be dedicated to this, which will be followed by the evaluation of the physical-mechanical properties suitable for the bone-integration application.

## Figures and Tables

**Figure 1 polymers-13-02751-f001:**
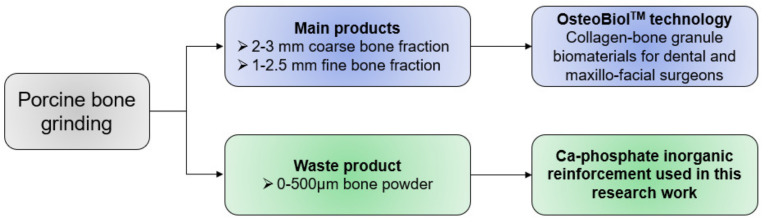
Origin of the BP (provided by Tecnoss™) used in this work.

**Figure 2 polymers-13-02751-f002:**
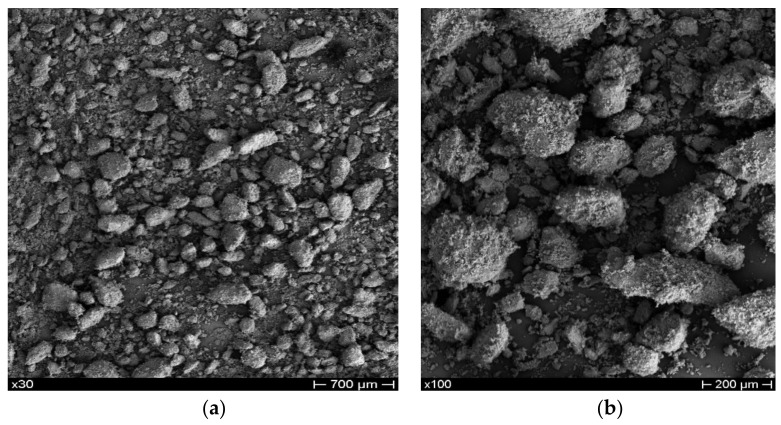
SEM images of BP at low (**a**) and high (**b**) magnifications.

**Figure 3 polymers-13-02751-f003:**
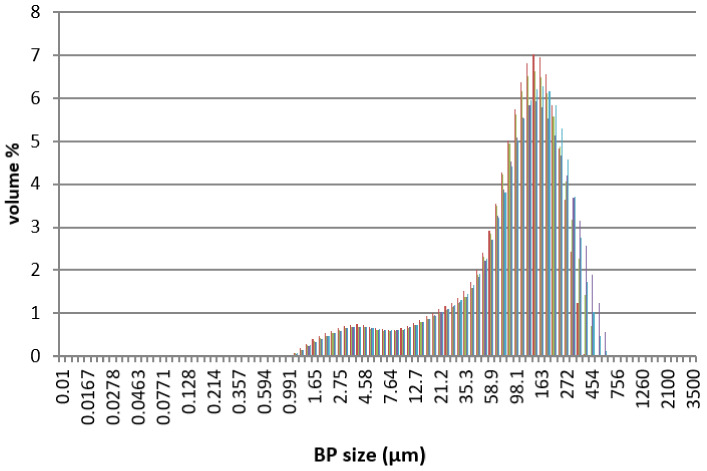
Dimensional distribution of the BP used in this work. Each color represents a particle size pair of BP (μm)–Volume (%) obtained from the MasterSizer analyzer.

**Figure 4 polymers-13-02751-f004:**
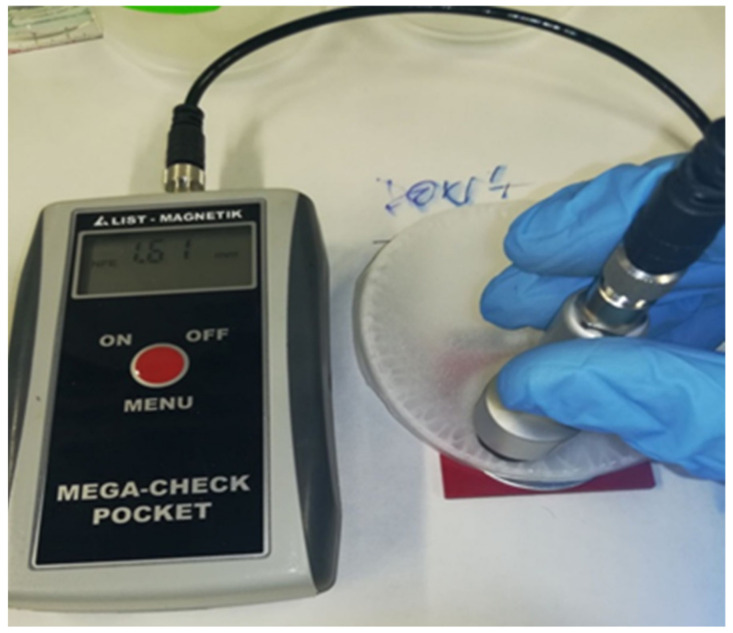
Thickness measurement on solvent-casted polymer-BP films.

**Figure 5 polymers-13-02751-f005:**
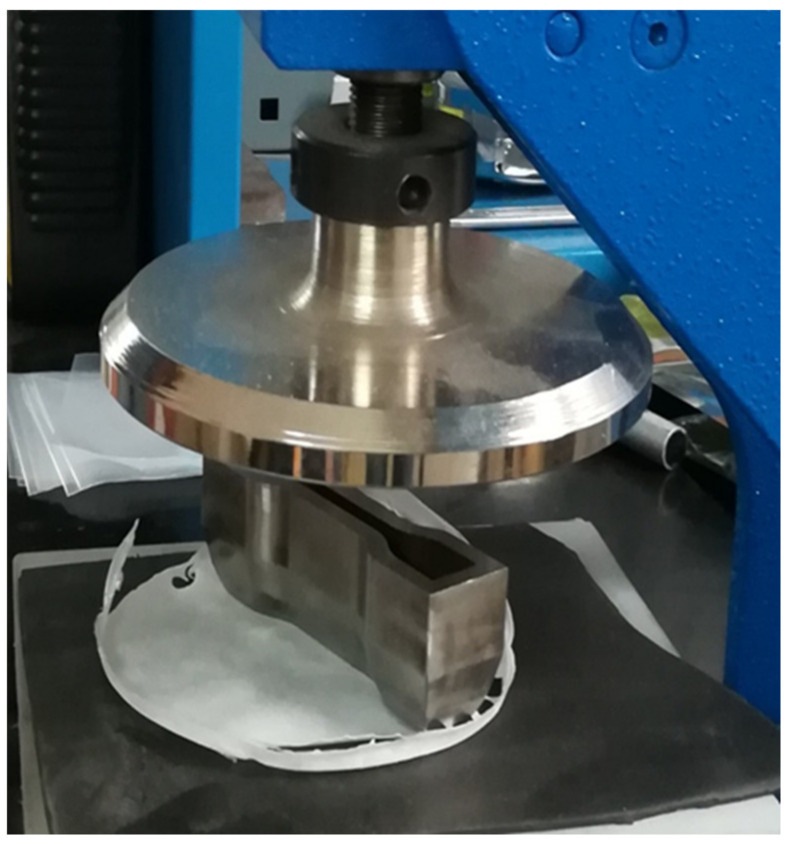
Dog-bone specimen preparation for tensile test: Die-cutting method.

**Figure 6 polymers-13-02751-f006:**
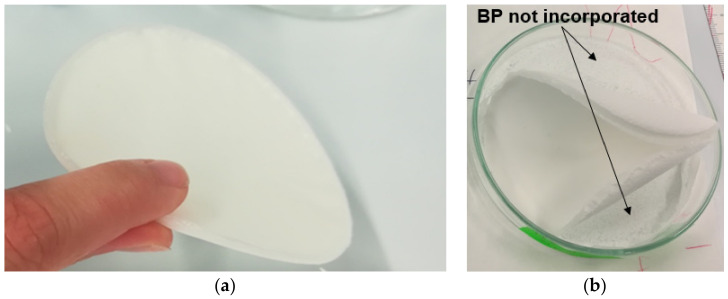
Comparison between a well-made disk (**a**) and a solvent-casted specimen with unsuitable features (**b**).

**Figure 7 polymers-13-02751-f007:**
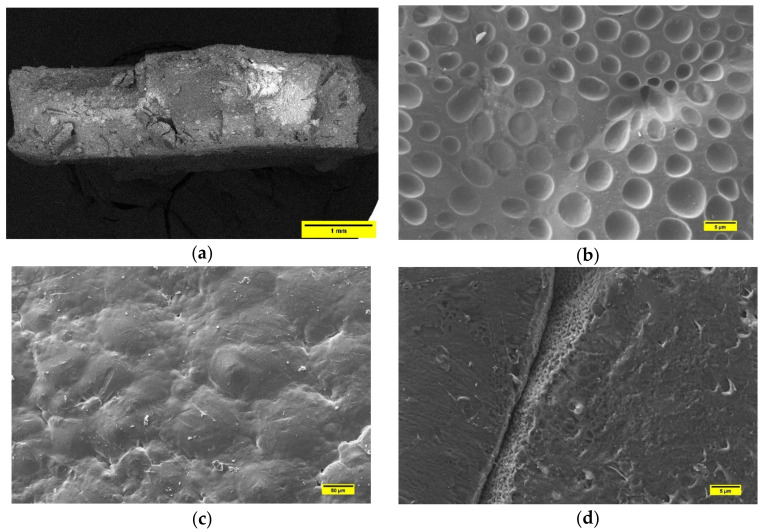
(**a**) Cross-section of PLA sample filled with 40 wt% of BP. (**b**) SEM micrograph PLA surface in the composite with 30 wt% of BP, highlighting air bubbles resulted from solvent casting; (**c**) SEM micrograph of PLA-BP sample (50 wt% of BP), where the inorganic particles are not detectable; (**d**) SEM image of sample constituted by 50 wt%BP and 50wt%PLA. In this case, the micrograph shows the poor cohesion between matrix and filler.

**Figure 8 polymers-13-02751-f008:**
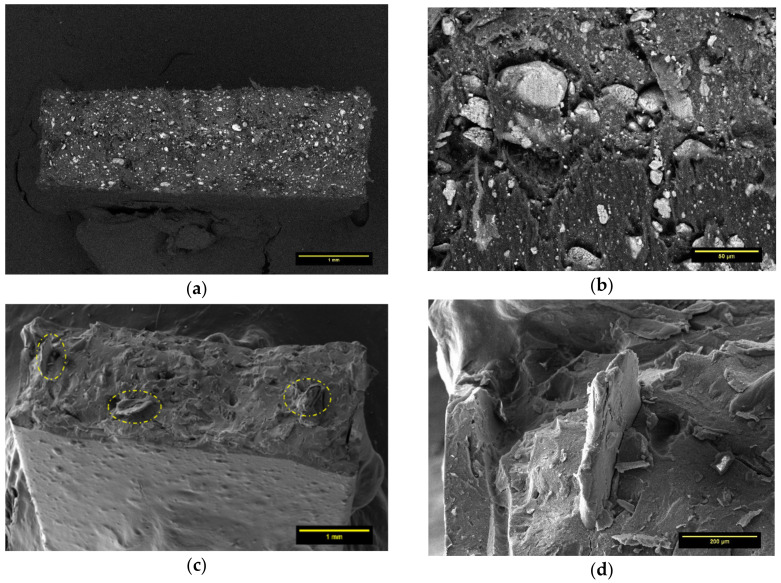
(**a**) Cross-section image of PCL50-BP50. (**b**) Interfacial adhesion between PCL and BP in PCL50-BP50 sample. (**c**) Distribution of CS filler in PCL45-BP50-CS5 specimen. (**d**) Detail on the interfacial adhesion of CS in PCL matrix.

**Figure 9 polymers-13-02751-f009:**
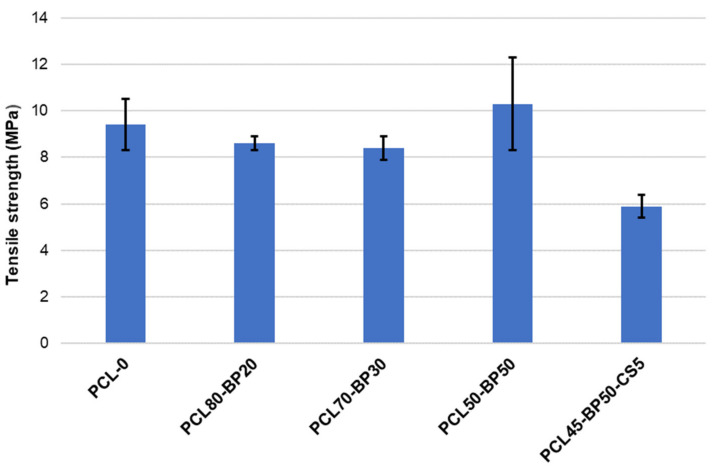
Tensile strength results.

**Figure 10 polymers-13-02751-f010:**
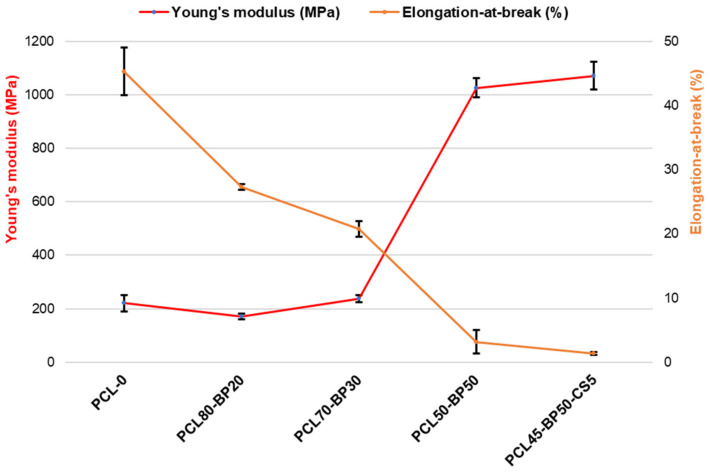
Young’s modulus and elongation-at-break results.

**Figure 11 polymers-13-02751-f011:**
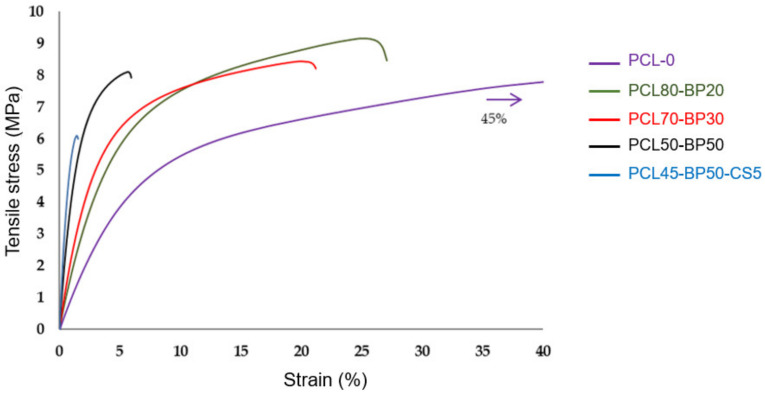
Stress vs. strain curves from tensile test.

**Figure 12 polymers-13-02751-f012:**
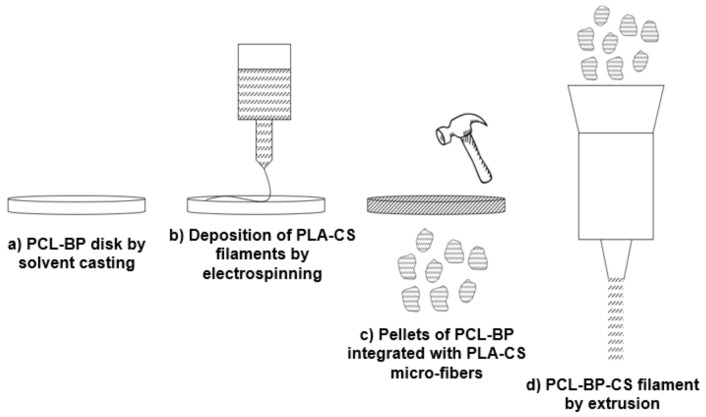
Implementation of the production technology of thermoplastic composites: Future upgrade.

**Figure 13 polymers-13-02751-f013:**
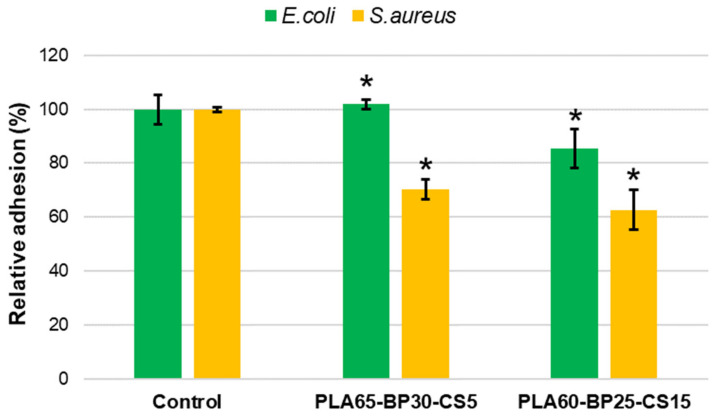
*S. aureus* and *E. coli* bacterial adhesion on PLA-CS-BP matrices. The inhibition of bacterial adhesion was significant with * *p* < 0.05 (*t*-Test).

**Figure 14 polymers-13-02751-f014:**
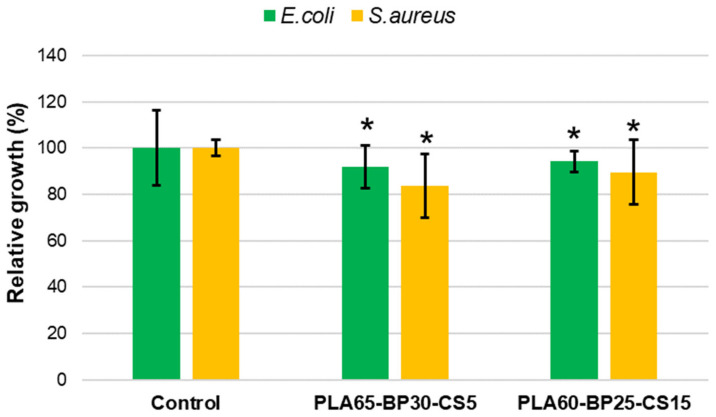
Relative growth of *S. aureus* and *E. coli* strains on CS loaded PLA-BP scaffolds after 24 h of culture. * *p* < 0.05 vs. control (*t*-Test).

**Table 1 polymers-13-02751-t001:** Advantages and drawbacks of the scaffold fabrication technologies for TE and comparison with transplantation methods.

Method	Advantages	Drawbacks
**Transplantation**		
Autografts	Availability of all necessary genetic elements (cells, tissue-inductive growth factors, substrates) for tissue regeneration [7]	Pain and morbidity of the donor site, prolonged surgery, limited available volume, and hardly manipulation to reproduce complex anatomical structures [7]
Allogeneic grafts	Virtually unlimited amount of material obtainable in various sizes and shapes from a human tissue bank. Shorter surgery compared to autografting because the tissue harvesting procedure is not necessary. Suitable healing capabilities [7,8]	Disease transmission, toxicity associated with the sterilization, high variability in host immune response, limited supplies. Lack of scientific evidence and standardized protocols [7,8]
**Conventional Fabrication Technique**		
Solvent casting	Low-cost technique. High porosity (50–90%). Not need of sophisticated equipment [6,9]	Use of very toxic solvents. Residual solvent can defunctionalize the cell growth [6,9]
Freeze-drying	Possibility to manage the pore size distribution by changing the freezing method. Possibility to produce scaffold with suitable interconnectivity without implementing high working temperature [6,9]	Use of cytotoxic solvents. Generation of small and irregular pores (15–35 μm) [6,9]
Gas foaming	High porosity (up to 85%). No use of organic solvents [6,9]	Strict control of thermal operating conditions is necessary as the method is highly sensitive to the development of closed pore structures and non-porous skin layer [6,9]
Electrospinning	Possibility to produce highly porous scaffolds with small pore diameters (nano to micro scale). High tensile strength performance due the fibers-based structure [6,9]	Use of toxic solvents. Process depends on many variables (type of solvent, polymer concentration, voltage, flow rate, needle size, temperature, pressure, needle-to-collector distance). Problematic to obtain 3D structures [6,9]
**Rapid Prototyping Methods**		
Stereolithography	High resolution. Uniformity in pores interconnectivity. Possibility to fabricate structures with anatomically shape [6,9]	Expensive equipment. Requiring massive amounts of monomers and post-treatment to improve the monomer conversion. Shrinkage during polymerization [6,9]
Selective laser sintering	Not using solvent. Rapid process. Excellent control over the scaffold microstructure by adapting the process parameters [6,9]	High operating temperature. High-cost equipment. Requiring many post-processing treatments to remove injected powder [6,9]
Fused deposition modelling	Low operating temperature. Solvent-free technique. Reaching of suitable strength properties. Low cost [6,9]	Medium precision. Several limitations in its application to biodegradable polymers [6,9]
Bioprinting	High accuracy and shape complexity. High cell viability (80–90%). Low costs [6]	Depends on existence of cells [6]

**Table 2 polymers-13-02751-t002:** Compositions of PCL-based samples characterized by tensile test.

ID Sample	PCL (wt%)	BP (wt%)	CS (wt%)
PCL-0	100	0	0
PCL80-BP20	80	20	0
PCL70-BP30	70	30	0
PCL50-BP50	50	50	0
PCL45-BP50-CS5	45	50	5

**Table 3 polymers-13-02751-t003:** Compositions of PLA-based samples for biological tests.

ID Sample	PLA (wt%)	BP (wt%)	CS (wt%)
PLA65-BP30-CS5	65	30	5
PLA60-BP25-CS15	60	25	15

## Data Availability

Not applicable.

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
