# Peer review of "Recycled Porcine Bone Powder as Filler in Thermoplastic Composite Materials Enriched with Chitosan for a Bone Scaffold Application"

_polymers, 2021, doi:10.3390/polym13162751_

Round 1

Reviewer 1 Report

This paper reports on preparation of a PCL/PLA based potential scaffold by solvent casting method. During the last decades scaffold preparation has been extensively studied by many research groups but authors did not take this opportunity. Although the paper has some merits, there are some deficiencies which make me rather uncertain in acceptance. Major points to be considered:

  • First of all the goal of the paper is unclear for me. PLA/PCL/chitosan composites are a well described and used system for 3D printing. I do not see any novelty in film casting experiments. The porcine bone powder would be the new finding? It should have been described somewhere.
  • The abstract is quite deceiving. Authors wrote about 3D printing and electrospinning but there are no experiments were carried about any of them but solvent casting. It is also mentioned here that they wanted to develop PCL-BP composites reinforced with PLA-CS microfibers. But this is also just a future plan because there is nothing about this “mixture” in the manuscript.
  • The introductory part does not support the rest of the paper; the number and quality of references are insufficient.
  • Bone powder is not a bio-ceramics. It is a Ca-phosphate. An inorganic material become a ceramic material if it undergoes some formation and sintering methods. (row: 75)
  • Bone-cement is not a ceramic material, it is polymer, PMMA (poly (methyl methacrylate). (row: 82)
  • What does eco-compatible mean? (row: 98)
  • Row: 158 – “The blends investigated…” – The materials in the paper are not blends, but composites.
  • In Table 1 there are composition data about composites but one can hardly find the same composition in order to compare the changes in the properties. There is no reference material in case of PLA, there is just one composition with PCL and CS. So it is impossible to compare something to something. In my opinion authors should not draw any conclusion if there are no measurements in a given (at least 5 different) composition range.
  • If there is a literature about the poor adhesion in PLA/BS composites why authors should repeat the experiments. (Row: 253)
  • In case of SEM images (Fig7.) to detect the backscattered electrons would be more informative because we could see the different elements and could distinguish the matrix and the filler.
  • I did not find any SEM images about chitosan filled composites.
  • Usually tensile strength and modulus changes equally with filler loading. How authors explain that while tensile strength decrease modulus increase?

Author Response

Polymerss-1306345

Cover letters for reviewer

Reviewer 1

The authors would like to thank the reviewer for his valuable comments and suggestions aimed at improving the present manuscript. Below are our replies to comments.

  1. First of all the goal of the paper is unclear for me. PLA/PCL/chitosan composites are a well described and used system for 3D printing. I do not see any novelty in film casting experiments. The porcine bone powder would be the new finding? It should have been described somewhere

1.r The novelty of this work concerns the attempt to incorporate an industrial waste product (i.e., porcine bone powder) as a bioactive filler for thermoplastic composites potentially applicable in the BTE sector. This aspect was emphasized from line 126 to line 132. Solvent casting was the technique used for a preliminary investigation. Future research will be aimed at studying more advanced methodologies to obtain the composites under examination (see Section 3.4).

  1. The abstract is quite deceiving. Authors wrote about 3D printing and electrospinning but there are no experiments were carried about any of them but solvent casting. It is also mentioned here that they wanted to develop PCL-BP composites reinforced with PLA-CS microfibers. But this is also just a future plan because there is nothing about this “mixture” in the manuscript.
  2. The introductory part does not support the rest of the paper; the number and quality of references are insufficient

3.r The introduction part it has been extensively modified and enhanced by the latest and highest quality reference (see the new references from [2] to [25])

  1. Bone powder is not a bio-ceramics. It is a Ca-phosphate. An inorganic material become a ceramic material if it undergoes some formation and sintering methods. (row: 75)

4.r “Bio-ceramic” term for BP has been replaced (with “inorganic filler”) or removed throughout the manuscript

  1. Bone-cement is not a ceramic material, it is polymer, PMMA (poly (methyl methacrylate). (row: 82)

5.r This part has been replaced with the new introduction based on the new references

  1. What does eco-compatible mean? (row: 98)

6r. “Eco-compatible” has been removed (see line 128). It was a mistake.

  1. Row: 158 – “The blends investigated…” – The materials in the paper are not blends, but composites.

7r. “Blend/blends” has been replaced with “composite/composites” throughout the manuscript

  1. In Table 1 there are composition data about composites but one can hardly find the same composition in order to compare the changes in the properties. There is no reference material in case of PLA, there is just one composition with PCL and CS. So it is impossible to compare something to something. In my opinion authors should not draw any conclusion if there are no measurements in a given (at least 5 different) composition range.

8r. In line 279, a text has been added to clarify that for both matrices (PCL and PLA) the same substitution levels of BP and CS were attempted. However, only some PCL formulations (Table 2) have been correctly obtained and characterized by tensile test. Some formulations in PLA (Table 3), on the other hand, have been investigated through antibacterial tests to evaluate the influence of two concentrations of CS on the biological performance of the composite, in accordance with the upgrade of section 3.2

  1. If there is a literature about the poor adhesion in PLA/BS composites why authors should repeat the experiments. (Row: 253)

9r. On this specific type of inorganic filler (recycled bone powder) there are no research works in literature. Therefore, it was worth evaluating its compatibility with the PLA matrix as well.

  1. In case of SEM images (Fig7.) to detect the backscattered electrons would be more informative because we could see the different elements and could distinguish the matrix and the filler.

10r. Thanks for the suggestion. We added figure 7a on BSE cross-section of PLA-BP composite

  1. I did not find any SEM images about chitosan filled composites.

11r. Figures 8c and 8d report SEM images of chitosan filled composites

  1. Usually tensile strength and modulus changes equally with filler loading. How authors explain that while tensile strength decrease modulus increase?

12r. In line 384 the authors report an explanation about this finding with proper reference. This behavior can be attributed to the lower PCL concentration than the other formulations. Similar finding was found by Fadaie et al. [30]: reducing concentration of PCL solution makes a remarkable decline in tensile strength.

Reviewer 2 Report

On request of Polymers, I have revised the manuscript titled “Recycled porcine bone powder as filler in thermoplastic composite materials enriched with chitosan for a bone scaffold application” by Marco Valente et al.

In this work, using recycled porcine bone powder (BP), known to be successful as bone substitute and bone reconstruction, the authors have synthesized and characterized biocompatible composite materials. Polylactic acid (PLA) and poly(ε-caprolactone) (PCL) were used as matrices and chitosan (CS) was included in the cocktail ingredients to confer to the prepared materials antibacterial properties. The solvent casting method has been (for the moment) investigated with the aim at introducing the highest possible percentage of BP and CS. According to the reported results the composites prepared possess mechanical properties suitable for hard tissue engineering applications, even if the antibacterial properties are questionable.

General comments.

Nowadays, the continuous advances in medical science and surgical techniques have allowed scientists to restore the native functions of many damaged parts of the human bod, by transplantation both of tissues and whole organs. Unfortunately, the extensive application of transplantation techniques is limited by the growing demand that far exceeds the actual availability by donors. To address this issue, tissue engineering was born as a branch of regenerative surgery and represents a new versatile approach for the repair/regeneration of damaged tissues, by using biomaterials and finds application mainly in orthopedic and dental sector. On this scenario, the development of an increasing number of new biomaterial increasingly improved is required urgently, and the herein study is desired for publication, as well as the results for the moment obtained are worthy of further investigations to be optimized.

Collectively, the present manuscript is well written, easy to read, and English is fine. This research will attract the attention of a wide audience among Polymers’ readers and not only. However, some part, mainly concerning the characterization of prepared materials and the evaluation of their antibacterial properties are improvable. My concerns and requests, which must be addressed before acceptance of this manuscript are listed below.

The abstract is extremely long. According to Polymers instructions, the abstract should contain at max 200 words. The herein reported abstract counts for 270 words. Please, shorten it accordingly.

The headings and sub-headings along all manuscript should be rewritten according to the format observable in the template (all in capital letters).

Along all manuscript, for all chemicals and instrument, the manufacturers, cities, and countries should be inserted. Correct along all manuscript where necessary.

In some Figure’s captions the final dot is missing. Please, correct captions, as necessary.

Please, in the Figures where it has not been used, use the Arial font. It is clearer.

In the Introduction, it would be fine to insert a Table collecting the main advantages and disadvantages associate to the transplantation techniques and tissue engineering approaches with the due references.

Lines 55-56. Badly written. Please, rewrite the sentence.

Line 166. Please, insert “by” before “a MEGA_CHECK”.

The authors should analyse PLA, PCL, CS, BS, and the bio-composites prepared by FTIR spectroscopy and compare the results by processing the spectral data (Matrix of data) by the principal components analysis (PCA), which is a chemometric tool extensively applied to well interpret FTIR results. The results must be reported and discussed in the proper Section.

Concerning the assay of bacterial growth and adhesion, the authors should be also including a representative species of Gram-negative bacteria, such as E. coli. The major activity of CS towards Gram-positive bacteria is not a rational justification for not having considered this clinically relevant bacterial family. In addition, experiments of bacterial growth and adhesion should be performed on pure CS used in this study for comparison.

I do not agree with what the authors have written at the beginning of Section 3.5. The antibacterial properties of low MW CS result higher or lower of those of high MW CS depending on the measure unit with which concentrations are expressed. If concentrations are expressed in µg/mL, high MW CS results usually less potent than low molecular weight CS. On the contrary, if the concentrations are expressed as µM, high MW CS results more potent than low molecular weight CS. Please, think about this and modify what you wrote accordingly.

Please, modify the author contribution section according to the Polymers template.

Since, I want to see the present manuscript after revision, I ask for major revisions.

Author Response

The authors would like to thank the reviewer for his valuable comments and suggestions aimed at improving the present manuscript. Below are our replies to comments.

  1. The abstract is extremely long. According to Polymers instructions, the abstract should contain at max 200 words. The herein reported abstract counts for 270 words. Please, shorten it accordingly.

1.r The abstract has been shortened in accordance with the reviewer's comment and the journal requirement

  1. The headings and sub-headings along all manuscript should be rewritten according to the format observable in the template (all in capital letters).
  2. Along all manuscript, for all chemicals and instrument, the manufacturers, cities, and countries should be inserted. Correct along all manuscript where necessary

3.r Manufacturers, cities, and countries have been inserted throughout the manuscript

  1. In some Figure’s captions the final dot is missing. Please, correct captions, as necessary.

4.r Done

  1. In the Introduction, it would be fine to insert a Table collecting the main advantages and disadvantages associate to the transplantation techniques and tissue engineering approaches with the due references.

5.r Table 1 has been added to reports a comparison between transplantation techniques and tissue engineering approaches in terms of pro and cons

  1. Lines 55-56. Badly written. Please, rewrite the sentence

6r. The sentence has been rewritten (see Line 76)

  1. Line 166. Please, insert “by” before “a MEGA_CHECK”.

7r. Done (see Line 197)

  1. The authors should analyse PLA, PCL, CS, BS, and the bio-composites prepared by FTIR spectroscopy and compare the results by processing the spectral data (Matrix of data) by the principal components analysis (PCA), which is a chemometric tool extensively applied to well interpret FTIR results. The results must be reported and discussed in the proper Section

8r. In this preliminary experimental campaign, it was not possible to plan FTIR analysis (also due to difficulties due to work restrictions due to the pandemic). We thank the reviewer for the comment and suggestion. It could be a valuable indication for future investigations on this topic.

  1. Concerning the assay of bacterial growth and adhesion, the authors should be also including a representative species of Gram-negative bacteria, such as E. coli. The major activity of CS towards Gram-positive bacteria is not a rational justification for not having considered this clinically relevant bacterial family. In addition, experiments of bacterial growth and adhesion should be performed on pure CS used in this study for comparison.

9r. Concerning the assay of bacterial growth and adhesion, we added the results on E. coli (see figures 13 and 14). However, the pure CS study could not be performed but will be implemented in future research. In this preliminary investigation, the aim was to evaluate the antibacterial effect of CS in the new composite developed by the authors.

  1. I do not agree with what the authors have written at the beginning of Section 3.5. The antibacterial properties of low MW CS result higher or lower of those of high MW CS depending on the measure unit with which concentrations are expressed. If concentrations are expressed in µg/mL, high MW CS results usually less potent than low molecular weight CS. On the contrary, if the concentrations are expressed as µM, high MW CS results more potent than low molecular weight CS. Please, think about this and modify what you wrote accordingly.

10r. Thanks a lot for your explanation. The begin of Section 3.5 (line 430-437) has been changed in accordance with the information provided by the reviewer and a supporting reference has been added (ref 33).

  1. Please, modify the author contribution section according to the Polymers template

11r. Done.

Reviewer 3 Report

The authors in this manuscript showed an interesting study that focuses on the use of thermoplastic composites incorporating recycled porcine bone powder along with chitosan to be used for the bone scaffold. In summary, the manuscript represents an interesting finding and is a good fit for Polymers journal. I believe that the quality of the article could be improved, provided the authors address some of the minor issues. I feel that the manuscript could be considered for publication after addressing the following queries/suggestions.
Some revisions which should be done are listed as follows:

(1) In the introduction section lines 55-56 the authors write "Therefore, different matrices have been identified technological and mechanical characteristics." This line is ambiguous and needs to be rephrased.

(2) Please rephrase the sentence "Many are the cases already made about these forms of applications" in lines 82-83 for better clarity and understanding.

(3) In lines 156-158, "In accordance with the above procedure, different formulations were tested by varying the type of matrix (PCL and PLA and the percentage of fillers (CS and BP)". Please remove the unaccounted round bracket.

(4) The authors mention "different formulations were tested by varying the type of matrix (PCL 157 and PLA and the percentage of fillers" On what basis did the authors choose the formulations?

(5) In table 1, Please provide the sample name along with the formulations for better understanding.

(6) Line 270-272, the authors write "From the cross-section image of 50 wt%BP-50 wt%PLA fragment, it can be seen how the polymer incorporates BP, which is homoge-neously distributed over the investigated section (Figure 8a)" However Figure 8a caption reads "Cross-section image of 50 wt%BP-50 wt% PCL". In text the authors talk about PLA matrix and in the figure caption the authors mention PCL matrix. Please correct this.

(7) For Mechanical characterization by tensile tests, Please provide Stress vs Strain curves for better understanding.

(8) In the Tensile testing results, the authors show error bars in the figures, however, no mention of how many samples were tested to obtain the standard deviation is provided. Please provide this information. 

(9) Why did the authors not study the antibacterial property of 50wt%BP-40wt%PCL-10wt%CS sample?

(10) The authors mention the"...to obtain 3D printable filaments..." in line 381 however no studies or characterizations have been done to support the 3D printability claim.

(11) Majority of the references (>50%) cited by the authors are almost older than 10 years. It would improve the impact of the article if more recent publications are cited.

Author Response

Polymerss-1306345

Cover letters for reviewer

Reviewer 3

The authors would like to thank the reviewer for his valuable comments and suggestions aimed at improving the present manuscript. Below are our replies to comments.

  1. In the introduction section lines 55-56 the authors write "Therefore, different matrices have been identified technological and mechanical characteristics." This line is ambiguous and needs to be rephrased

1.r The sentence has been rephrased (Line 76)

  1. Please rephrase the sentence "Many are the cases already made about these forms of applications" in lines 82-83 for better clarity and understanding
  2.  In lines 156-158, "In accordance with the above procedure, different formulations were tested by varying the type of matrix (PCL and PLA and the percentage of fillers (CS and BP)". Please remove the unaccounted round bracket.

3.r Done

  1. The authors mention "different formulations were tested by varying the type of matrix (PCL 157 and PLA and the percentage of fillers" On what basis did the authors choose the formulations?

4.r In line 279, a text has been added to clarify that for both matrices (PCL and PLA) the same substitution levels of BP and CS were attempted. However, only some PCL formulations (Table 2) have been correctly obtained and characterized by tensile test. Some formulations in PLA (Table 3), on the other hand, have been investigated through antibacterial tests to evaluate the influence of two concentrations of CS on the biological performance of the composite, in accordance with the upgrade of section 3.2

  1. In table 1, Please provide the sample name along with the formulations for better understanding

5.r Tables 2 and 3 report the sample name

  1. Line 270-272, the authors write "From the cross-section image of 50 wt%BP-50 wt%PLA fragment, it can be seen how the polymer incorporates BP, which is homoge-neously distributed over the investigated section (Figure 8a)" However Figure 8a caption reads "Cross-section image of 50 wt%BP-50 wt% PCL". In text the authors talk about PLA matrix and in the figure caption the authors mention PCL matrix. Please correct this

6r. Correct (Line 328)

  1. For Mechanical characterization by tensile tests, Please provide Stress vs Strain curves for better understanding

7r. Figure 11 reporting stress vs strain curves has been added.

  1. In the Tensile testing results, the authors show error bars in the figures, however, no mention of how many samples were tested to obtain the standard deviation is provided. Please provide this information

8r. The information about the number of samples tested in tensile is reported in Line 210

  1. Why did the authors not study the antibacterial property of 50wt%BP-40wt%PCL-10wt%CS sample?

9r. Because, according to the proposal reported in Section 3.2, we were interested in verifying the effect of CS only in PLA composites. The investigated formulations, for clarity, are shown in Table 3

  1. The authors mention the"...to obtain 3D printable filaments..." in line 381 however no studies or characterizations have been done to support the 3D printability claim.

10r. According to the reviewer’s comment, we changed the sentence removing 3D printing for better fit with the paper’s aim (see Line 466)

  1. Majority of the references (>50%) cited by the authors are almost older than 10 years. It would improve the impact of the article if more recent publications are cited.

11r. More recent and higher quality references have been added, mainly in the introduction part, which has been modified accordingly.

Round 2

Reviewer 1 Report

Accept in present form

Author Response

Thank you very much for the time.

Regards

Marco Valente

Reviewer 2 Report

I have revised the modified version of the herein manuscript provided by the authors. Following almost all my suggestions the authors have significantly improved their paper, but further minor revisions are necessary before publication.

1) All the words of headings and sub-headings must be written with the first letter in capital form.

2) I think that to investigate the antibacterial activity of CS alone is essential to verify the utility of the study. On the other hand I understand that the authors could have difficulties in performing further experiments, as in the case of FTIR analyses that I had requested. So, I ask the authors to search for litterature reported data concerning the antibacterial activity of CS with MW like that used by them in this study and to add this data in the manuscript. Then, after having compared such data with that of their materials a discussion will be necessary.  

Author Response

Polymers-1306345

Cover letters for reviewer (2nd round)

Reviewer 2

The authors thank the reviewer for the further comments and suggestions aimed at improving the manuscript. Below our response to the comments:

  1. All the words of headings and sub-headings must be written with the first letter in capital form.

1r. Done

  1. I think that to investigate the antibacterial activity of CS alone is essential to verify the utility of the study. On the other hand I understand that the authors could have difficulties in performing further experiments, as in the case of FTIR analyses that I had requested. So, I ask the authors to search for litterature reported data concerning the antibacterial activity of CS with MW like that used by them in this study and to add this data in the manuscript. Then, after having compared such data with that of their materials a discussion will be necessary.

2.r  Thanks for your understanding and for your valuable suggestion. Literature data on the antibacterial activity of chitosan have been added (like the one investigated in this work) in line 437. In addition, a brief discussion of the results has been added in line 460. On the new part have been added three new references [34][35][39].